# Impact of Shiftwork on Retinal Vasculature Diameters over a 5-Year Period: A Preliminary Investigation Using the BCOPS Study Data

**DOI:** 10.3390/ijerph21040439

**Published:** 2024-04-03

**Authors:** Luenda E. Charles, Ja K. Gu, John M. Violanti

**Affiliations:** 1Health Effects Laboratory Division, National Institute for Occupational Safety and Health, Centers for Disease Control and Prevention, Morgantown, WV 26505-2888, USA; gum4@cdc.gov; 2Department of Epidemiology and Environmental Health, School of Public Health and Health Professions, State University of New York at Buffalo, Buffalo, NY 14214-8001, USA; violanti@buffalo.edu

**Keywords:** CRAE, CRVE, retinal arterioles, retinal venules, shiftwork, police officers

## Abstract

Our aim was to investigate the impact of shiftwork on changes in central retinal arteriolar equivalent (CRAE), a measure of arteriolar width, and central retinal venular equivalent (CRVE), a measure of venular width, over five years. The participants were 117 officers (72.7% men) examined at the first (2011–2014) and second (2015–2019) follow-up examinations in the Buffalo Cardio-Metabolic Occupational Police Stress study. Shiftwork data were obtained from the City of Buffalo, NY payroll records. Retinal diameters were measured using a standardized protocol. ANCOVA was used to compare mean change in CRAE and CRVE between the two examinations across shiftwork categories. Among men only, those who worked ≥70% hours on day shifts had a larger decrease in mean CRAE (−7.13 µm ± 2.51) compared to those who worked <70% day (−0.08 ± 0.96; *p* = 0.011). Among patrol officers, those who worked ≥70% day had a larger decrease in CRAE compared to those who worked <70% day (*p* = 0.015). Also, officers who worked ≥70% day had an increase in mean CRVE (µm) (4.56 ± 2.56) compared to those who worked <70% (−2.32 ± 1.32; *p* = 0.027). Over the five-year period, we observed adverse changes in arteriolar and venular diameters among officers who worked ≥70% on day shifts. The results should be interpreted with caution due to the small sample sizes.

## 1. Introduction

Shiftwork has been defined as work hours outside the daylight hours of 7:00 a.m. to 6:00 p.m. and includes fixed evening or night shifts, rotating shifts, and weekend shifts [1]. Shiftwork is necessary in occupations where services must be provided on a 24-h basis, and the prevalence has increased over decades as other businesses feel the need to provide continuous assistance [2,3]. The benefits that shiftwork brings for employers, employees, and the public at large are tempered by numerous health problems associated with shiftwork. The International Agency for Research on Cancer (IARC) has identified night shift work as a probable human carcinogen [4]. Shiftwork has also been shown to be associated with poorer sleep quality, sleep deprivation, hypertension, insulin resistance and diabetes, obesity, metabolic syndrome, depression and other mental health problems, cerebrovascular disease, ischemic heart disease, and coronary heart disease [5,6,7,8,9,10,11].

Coronary heart disease and other diseases affecting the macrovascular system have been shown to be associated with microvascular abnormalities. In one study from Denmark, the results showed that individuals with narrower retinal arterioles were more likely to have diabetic nephropathy and macrovascular disease [12]. In a cross-sectional study of 173 hypertensive patients, the authors found an inverse and significant association between carotid intima media thickness (CIMT) and retinal arteriolar caliber (β = −0.245, *p* = 0.001) and positive significant associations between CIMT and retinal venular caliber (β = 0.191, *p* = 0.009) after controlling for age, gender, systolic blood pressure (SBP), total cholesterol, high-density lipoprotein (HDL) cholesterol, prior CVD, carotid plaque, and the retinal fellow vessel [13]. In yet another study, the authors compared retinal microvascular function in healthy individuals with and without a positive family history of CVD [14]. Among participants with a positive CVD family history, macrovascular function appeared intact, although with a low Framingham Risk Score (a tool designed to estimate heart disease risk based on data from the Framingham Heart Study). However, the authors reported that the retinal microvasculature had functional impairments that correlated with established plasma markers for cardiovascular risk. Some investigators have described the retinal microvasculature as a window to the heart [15,16]. Tedeschi-Reiner and colleagues (2005) explored the relationship between atherosclerosis of the retinal arteries and the extent and severity of coronary artery disease in 109 patients aged 40 to 80 years. The results showed that the extent and severity of retinal vessel atherosclerosis were strongly correlated with the extent and severity of coronary artery disease (CAD). Other studies have shown that arteriolar narrowing and venular widening precede the development of several health conditions [17,18,19,20,21]. These conditions include incident diabetes mellitus, incident heart failure, incident lacunar stroke, incidence and progression of diabetic retinopathy, and progression of cerebral small vessel disease. 

Shiftwork is an integral part of the occupation of all first responders, including law enforcement officers. There is ample evidence of the relationship between shiftwork and macrovascular disease but much less evidence of the relationship with microvascular disease and even less evidence regarding whether shiftwork might be a risk factor for microvascular problems. Therefore, our objective was to investigate whether shiftwork is an independent risk factor for adverse changes of central retinal arteriolar equivalent (CRAE), a measure of retinal arteriolar width, and central retinal venular equivalents (CRVE), a measure of retinal venular width, in police officers. A decrease in CRAE and an increase in CRVE both signify worse retinal health. A decrease in CRAE means a narrowing in retinal arteriolar width which is associated with health problems. In contrast, an increase in CRVE means a widening in retinal venular width and is known to be associated with health problems. We hypothesized that officers who worked on the afternoon and/or night shifts or those who worked a smaller portion of the time on the day shift would show a greater adverse impact on their retinal arteriolar and venular widths compared to those who worked a greater portion on the day shift.

## 2. Materials and Methods

### 2.1. Study Design and Participants

The participants of this study are police officers who were recruited to take part in the Buffalo Cardio-Metabolic Occupational Police Stress (BCOPS) study. The goal of the BCOPS study was to investigate associations between stressful occupational exposures and subclinical measures of CVD [22]. This study was reviewed and approved by the University of Buffalo Institutional Review Board.§ The baseline examination was conducted from June 2004 to October 2009, where 464 active-duty and retired police officers from approximately 710 officers were examined. Female officers who were pregnant at the time of examination were excluded (n = 2). The officers reviewed and signed informed consent forms before the examinations. Data for all exams were collected at the Center for Health Research, School of Public Health and Health Professions, University at Buffalo, State University of New York [22]. 

Some of the officers also participated in subsequent examinations: a first follow-up examination (2011–2014) and a second follow-up examination (2015–2019). Retinal photography data were not available during the baseline examination; they were only collected during these two follow-up examinations. During the first follow-up, 300 officers participated after signing consent forms (Figure 1). From this number, we removed 14 officers who had retired and 110 officers who did not participate in the second follow-up examination, resulting in 176 officers. We further removed 59 officers: those who had missing data on shiftwork (n = 2) and on retinal photographs (n = 28) and those who reported being diagnosed with cataracts, glaucoma, or ocular injuries (n = 29). The final sample size for this analysis included 117 police officers, including 32 women and 85 men.

### 2.2. Assessment of Shiftwork

We obtained electronic work history data from the City of Buffalo, NY payroll records, which were available for each day from May 1994 to the date of each officer’s second retinal photographic examination (2015–2019). The database contained information regarding the activities of each officer and included the start and end time of work, the type of activity (i.e., regular work, overtime work, and court appearances), the type of leave (i.e., weekend, vacation, work-related injury, and other types of sick leave), and the number of hours spent on each activity. All officers were scheduled to work four days on and three days off, with 10-h permanent, non-rotating shifts. The time officers started their shift for the regular time work was used to classify each record into one of the following three shifts: day shift if the start time of the record was between 0400 and 1159; afternoon shift if the start time was between 1200 and 1959; and night shift if the start time was between 2000 and 0359. An officer’s dominant shift was defined as the shift on which he/she worked the highest percentage of hours. For example, the dominant shift would be the night shift for an officer who worked 10% on the day shift, 5% on the afternoon shift, and 85% on the night shift. Dominant shift was defined for two time periods: across career and during the past one-year period prior to the clinical examination. We also created another variable, the percentage of hours worked on the day shift (≥70% vs. <70%) across career, to be used as an exposure variable. The 70% cut-point was selected as the best point to separate those who predominantly worked on the day shift versus other shifts. There is no biological justification for this cut-point. Lowering the 70% cut-point to 60% will increase the sample size but will also introduce participants who worked nearly 40% on afternoon or night shifts into this category.

### 2.3. Assessment of Retinal Vessel Diameters

Research associates from the University at Buffalo SUNY, NY were trained and certified to perform retinal imaging. Retinal imaging is a simple, non-invasive technology for assessing microvascular abnormalities that may be associated with CVD development. A non-mydriatic ophthalmic digital imaging system was used to take two digital images per eye (four total) on each imaging visit through a non-pharmacologically dilated pupil. Participants were seated in a windowless room with the lights turned off to allow the pupils to dilate naturally in preparation for the retinal imaging examination. One image was centered on the macula and the second on the optic nerve. The digital images were sent to the University of Wisconsin Department of Ophthalmology Ocular Epidemiology Reading Center to be graded using a standardized protocol. Retinal vessel diameters were measured at the Reading Center using a computer-assisted technique based on a standard protocol and using the Parr-Hubbard-Knudtson formula [23,24]. Trained graders, masked to participant characteristics, using a computer software program, measured the diameters of all arterioles and venules coursing through a specified area one-half to one disc diameter surrounding the optic disc. On average, between 7 and 14 arterioles and an equal number of venules were measured per eye. Individual arteriolar and venular measurements were combined into summary indices that reflected the average CRAE and CRVE diameters of an eye based on the Parr-Hubbard-Knudtson formula [25,26].

### 2.4. Covariates

Demographic characteristics, lifestyle behaviors, medical history, and medication use were obtained from all officers through self- and interviewer-administered questionnaires at the first follow-up examination. Smoking status was categorized as current, former, and never. Body mass index (BMI) was calculated as weight (in kilograms) divided by height (in meters) squared. Waist circumference was measured twice at the midpoint between the lowest rib and the top point of the hip bone, and the average value was used in the analysis. Blood pressure was determined using the average of the second and third of three separate measurements of resting systolic and diastolic blood pressure obtained with a standard sphygmomanometer. Hypertension was defined as a systolic blood pressure of ≥140 mmHg or a diastolic blood pressure of ≥90 mm Hg or self-reported use of antihypertensive medications (https://www.heart.org/en/health-topics/high-blood-pressure, accessed on 23 March 2023). White blood cell (WBC) count was obtained from a complete blood count using standard laboratory procedures. Sleep quality and sleep duration were assessed using the Pittsburgh Sleep Quality Index (PSQI) questionnaire [27].

### 2.5. Statistical Analysis

Descriptive statistics were obtained for all variables using the chi-square test of independence and the analysis of variance (ANOVA). Associations between selected variables and shiftwork were obtained using ANOVA and the chi-square test. There were strong correlations between the left and right eyes for the arteriolar diameters and the venular diameters; therefore, we used the average of the two eyes in our analyses, as has been conducted in other large studies [28,29]. We used three shiftwork variables at the first follow-up exam (2011–2014) as predictors: dominant shift during the entire career, dominant shift in the past year, and the percentage of hours worked on the day shift during the entire career. The two outcome variables were changes in CRAE and CRVE (values from the 2nd follow-up examination minus that from the 1st follow-up examination). All predictors were obtained from the 1st follow-up examination. Analysis of covariance (ANCOVA) was used to compare mean change in the outcome variables across categories of shiftwork, adjusting for covariates. To select potential confounders for adjustment in the models, we selected variables that were significantly associated with both the exposure (shiftwork) and the outcomes (change in CRAE and CRVE) but were not on the causal pathway (i.e., mediators), while adjusting for age. Confounders included age, sex, and race/ethnicity. In the final models, we also included certain risk factors for atherosclerosis: hypertension, BMI, and waist circumference for models of CRAE and WBC count and sleep duration for models of CRVE. WBC count was shown to be associated with wider CRVE [30]. Other risk factors for atherosclerosis (e.g., diabetes, total cholesterol, HDL and LDL cholesterol, and triglycerides) were not significantly associated with both the exposure and outcomes; therefore, they did not qualify as confounders and were not included in the models. We followed the pattern of adjusting for the change in CRAE (or CRVE) in all final models, as has been conducted in previous studies where investigators controlled for CRAE/CRVE in retinal diameter studies [21]. We stratified the analysis by sex and police rank. Due to inherent biological differences, stratification by sex is usually performed in epidemiological studies even when tests for interaction are not statistically significant. We stratified by police rank mainly because of the different roles assigned to the various ranks; rank could be considered a surrogate for different exposures. Statistical significance was indicated if the *p*-value was <0.05. All analyses were conducted in SAS v9.4 (SAS Institute, Cary, NC, USA).

## 3. Results

### 3.1. Study Sample Characteristics

These analyses included the 117 officers who participated in both the first and second follow-up examinations. In Table 1, we present descriptive statistics of officers collected at the first follow-up examination across shiftwork status based on their full career as an officer with the Buffalo Police Department. The mean (±SD) age of the police officers was 44.4 ± 7.7 years; ages ranged from 26 to 65 years, and 27.4% of the sample was women. The majority of officers were White (82.1%) and had a rank of patrol officer (59.0%). Sex was significantly associated with shiftwork (*p* = 0.006); a higher percentage of women was scheduled on the day shift compared to afternoon or night shifts. The workload of the officers was not significantly associated with shiftwork status, but having a second job was significantly associated with shiftwork status (*p* = 0.023). 

### 3.2. Shiftwork and CRAE

Table 2 shows the impact of shiftwork on change in CRAE. The dominant shift worked across career and past year did not significantly impact the 5-year change in CRAE, that is, the mean change in CRAE did not differ significantly across the shiftwork categories (day, afternoon, and night) for the two time periods (across career or past year). However, officers who, during their entire police career, worked ≥70% of the time on the day shift had a much larger significant mean decrease in CRAE (−5.57 µm ± 1.71) compared to those who worked <70% on the day shift (0.56 ± 0.89; *p* = 0.003) after adjustment for age, sex, hypertension, BMI, waist circumference, and change in CRVE between the two examinations.

Sex- and rank-stratified results of the impact of shiftwork on change in CRAE are presented in Table 3. Among female officers, we observed larger decreases in mean CRAE among those who worked the day shift compared to those who worked the afternoon/night shifts. Women who worked the day shift during the past year had a decrease in mean CRAE of −3.09 ± 1.58, while those who worked the afternoon/night shifts (during the same period) had an increase in mean CRAE (3.83 ± 2.79; *p* = 0.045), but the association was attenuated after adjustment for age, hypertension, BMI, waist circumference, and change in CRVE (*p* = 0.125). Among men, mean CRAE decreased in both groups (day = −0.92 ± 1.50 and afternoon/night = −0.77 ± 1.25), and the association was not statistically significant (multivariable-adjusted, *p* = 0.939).

Among male officers, those who worked ≥70% on the day shift had a much larger decrease in mean CRAE (−7.13 ± 2.51) compared to those who worked <70% on the day shift (−0.08 ± 0.96, *p* = 0.011) after multivariable adjustment (Table 3). Female officers who worked ≥70% on the day shift also showed a large decrease in mean CRAE (−3.00 ± 2.02) compared to those who worked <70% on the same shift (2.02 ± 2.02), but the result was not statistically significant (*p* = 0.099). 

Patrol officers who worked ≥70% on the day shift had a decrease in mean CRAE of −6.75 ± 2.45 compared to those who worked <70% on the day shift (0.48 ± 1.25, *p* = 0.015) after multivariable adjustment (Table 3). The changes in mean CRAE among the non-patrol officers followed a similar pattern (≥70% on day shift = −4.73 ± 2.61 vs. <70% = 0.92 ± 1.33), although the results were not statistically significant (*p* = 0.072).

### 3.3. Shiftwork and CRVE

Table 4 shows the impact of shiftwork on change in CRVE. Officers who, over their entire career, worked the afternoon shift had an increase in mean CRVE (2.92 µm ± 1.39) compared to those who worked the day shift (−1.00 ± 1.36; *p* = 0.051) and had a significant increase compared to those who worked the night shift (−1.46 ± 1.62; *p* = 0.041) after adjustment for age, sex, WBC count, sleep duration, and change in CRAE between the two examinations. Also, officers who worked ≥70% on the day shift had an increase in mean CRVE (3.15 ± 1.84) compared to those who worked <70% on the same shift (−0.70 ± 0.95) after multivariable adjustment, although not statistically significant (*p* = 0.078).

We stratified the impact of shiftwork on change in CRVE by sex and police rank (Table 5). Among women, those scheduled on the day shift in the past year had a significant increase in CRVE (3.66 ± 1.34) compared to those scheduled on the afternoon/night shift (−3.45 ± 2.70; *p* = 0.037). We did not observe any significant differences among men. The mean change in CRVE during the 5-year period was not significantly different across shiftwork among non-patrol officers. However, among patrol officers, those who worked ≥70% on the day shift had an increase in mean change of CRVE (4.56 ± 2.56) whereas those who worked <70% on the day shift had a decrease in mean change of CRVE (−2.32 ± 1.32) after multivariable adjustment (*p* = 0.027).

## 4. Discussion

This longitudinal study was initiated to investigate the effect of shiftwork on changes in the widths of the retinal arterioles and venules among police officers. As people age, structural changes are known to take place within the retinal arterioles and venules (i.e., arteriolar narrowing and venular widening) [17]. These changes become more pronounced over several years, but changes may also be observed over a period as short as five years, hence the reason for this study. The results from the Beaver Dam Eye study showed that men had larger CRVE than women but the change in CRVE over time was not significantly different from that of women [31]. The article by Myers and colleagues did not investigate retinal diameter changes in relation to shiftwork status. In fact, to the best of our knowledge, there were no studies that investigated the effect of shiftwork status on retinal diameters. 

We had expected to find more adverse changes in the arterioles and venules among night and/or afternoon shift officers compared to day shift officers. Our findings were different from what we had expected. Previous studies of shiftwork have shown that shift workers (night and/or afternoon) were the ones to experience adverse outcomes or no significant difference in outcomes. However, we observed that working ≥70% of the time on the day shift was a significant risk factor for adverse changes in the diameters of the arterioles and venules. Preliminary analyses of our data did not find significant differences between stress, physical activity, or workload among officers working different shift schedules. Additional studies would be required to elucidate the underlying reasons for the different results observed between these groups of officers.

### 4.1. Shiftwork and Arteriolar Diameters

Officers who worked ≥70% of the time on the day shift had a much larger significant decrease in mean CRAE (i.e., narrower arterioles) compared to those who worked <70% on the day shift. When stratified by sex, male officers who worked ≥70% on the day shift had a much larger decrease in arteriolar diameters compared to those who worked <70% on the day shift. Female officers who worked ≥70% on the day shift also showed a decrease in arteriolar diameter but the result was not statistically significant. When stratified by police rank, patrol officers who worked ≥70% on the day shift also had a significant decrease in arteriolar diameter compared to those who worked <70% on the day shift. However, we did not observe such significant associations among the non-patrol officers. 

### 4.2. Shiftwork and Venular Diameters

Officers who worked the afternoon shift for their entire career had a significant increase in venular diameter compared to those who worked the day shift or those who worked the night shift. Our results showed that officers who worked ≥70% on the day shift had an increase in venular diameter compared to those who worked <70% on the same shift, although this result did not reach statistical significance. After stratification by police rank, venular diameter significantly increased among patrol officers who worked ≥70% on the day shift compared to those who worked <70% but not among non-patrol officers. 

This study is important because changes that take place in the retinal microvessels usually predict what will happen in the macrovascular system [17,18,19,20,21]. This investigation of risk factors for microvascular dysfunction contributes to our understanding of risk factors that may later result in diseases of the macrovascular system. Abundant evidence is available showing the importance of maintaining health in the microvessels as a means of overall good health [32]. Narrower retinal arterioles are associated with worse health outcomes, e.g., chronic kidney disease, diabetes mellitus, hypertension, left ventricular hypertrophy, stroke, and coronary heart disease [17,20,21,33,34,35]. Wider retinal venules are associated with worse health outcomes, e.g., systemic inflammation, metabolic syndrome, endothelial dysfunction, heart failure, atherosclerosis, and stroke [17,21].

### 4.3. Limitations and Strengths

One limitation of our study is the relatively small sample size, which reduces the power for additional stratified analysis. Another limitation is the fact that the participants were members of one police department in the northeastern US, which limits the generalizability of our findings. A third limitation is that there is no biological justification for using 70% as the cut-point. We arbitrarily chose the 70% cut-point as the best choice to separate those who worked a large portion of the time on the day shift from other shifts. Also, as is sometimes the case with observational research, there may be unmeasured confounding variables that were not taken into consideration in our analyses. We cannot say with any certainty how the inclusion of such variables would have affected our findings. However, this study has a few strengths that are worth mentioning. We used objective shiftwork data, obtained from the City of Buffalo, NY payroll records. This is a major strength as many of the studies that investigate shiftwork as exposure only have access to self-reported data. Another strength is that, to the best of our knowledge, this is the first study to investigate whether shiftwork is a risk factor for adverse changes in CRAE and CRVE. We were unable to identify other published studies that investigated the impact of shiftwork on retinal arterioles and venules.

## 5. Conclusions

The results of our study are surprising. Working ≥70% on the day shift was significantly associated with narrower (i.e., worse) retinal arterioles among male officers and those who worked as patrol officers and wider (i.e., worse) venules, primarily among patrol officers. Night shift work has been typically associated with the worst health outcomes of all types. These results should be interpreted with caution due to the small sample sizes. This research topic warrants further investigation to extricate the underlying reasons for worse microvascular health among police officers scheduled mostly on the day shift. Future studies should employ a larger sample of workers and include more women.

## Figures and Tables

**Figure 1 ijerph-21-00439-f001:**
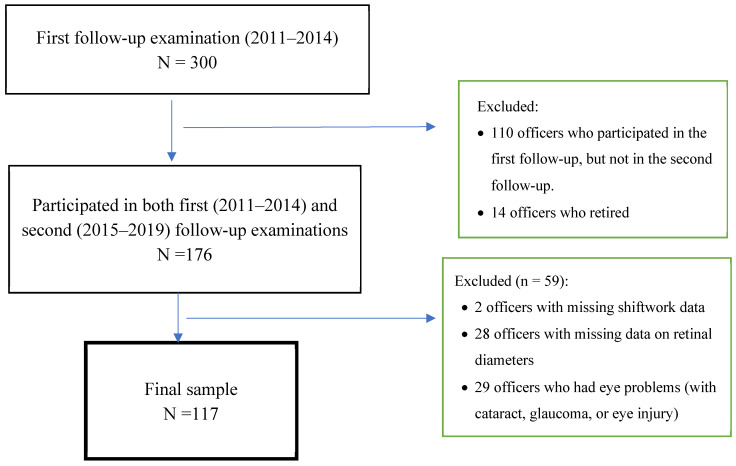
Flow chart showing exclusion and inclusion criteria and sample sizes.

**Table 1 ijerph-21-00439-t001:** Descriptive statistics of the study participants by shiftwork (entire career): BCOPS study, 1st follow-up examination (2011–2014).

	All (n = 117)	Day (n = 46)	Afternoon (n = 40)	Night (n = 31)	*p*-Value
	Mean ± SD	Mean ± SD	Mean ± SD	Mean ± SD	
Age (range = 26–65 years)	44.4 ± 7.7	46.4 ± 8.8	42.7 ± 6.8	43.7 ± 6.8	0.068
Years of service	17.0 ± 8.2	19.2 ± 9.6	16.0 ± 7.2	15.2 ± 6.5	0.064
Physical activity (h/wk)	6.9 ± 8.1	6.2 ± 8.6	7.4 ± 5.9	7.4 ± 9.7	0.769
Alcohol intake (drinks/wk)	4.2 ± 7.1	6.7 ± 7.0	5.2 ± 7.4	3.8 ± 6.9	0.580
Systolic blood pressure (mmHg)	114.0 ± 10.5	115.6 ± 10.6	113.0 ± 11.1	113.2 ± 9.8	0.463
Diastolic blood pressure (mmHg)	77.2 ± 7.7	78.2 ± 7.9	76.0 ± 8.0	77.4 ± 6.9	0.387
Body mass index (kg/m^2^)	28.4 ± 4.2	28.1 ± 4.1	28.7 ± 5.0	28.4 ± 3.3	0.837
Percent body fat (%)	25.8 ± 6.2	27.2 ± 6.0	25.2 ± 6.6	24.2 ± 5.6	0.099
Waist circumference (cm)	94.2 ± 12.8	92.8 ± 13.5	95.3 ± 13.6	94.9 ± 10.9	0.627
Sleep duration (h/24-h)	6.1 ± 1.1	6.3 ± 1.1	6.2 ± 1.2	5.7 ± 1.1	0.099
White blood cell (WBC) count (×10^9^/L)	5.7 ± 1.6	5.4 ± 1.3	5.7 ± 1.3	6.2 ± 2.3	0.123
	**N (%)**	**N (%)**	**N (%)**	**N (%)**	
Sex Women Men	32 (27.4) 85 (72.7)	20 (43.5) 26 (56.5)	8 (20.0) 32 (80.0)	4 (12.9) 27 (87.1)	0.006
Race/ethnicity White African American Hispanic	96 (82.1) 19 (16.2) 2 (1.7)	32 (69.6) 14 (30.4) 0 (0.0)	39 (97.5) 0 (0.0) 1 (2.5)	25 (80.7) 5 (16.1) 1 (3.2)	<0.001
Education ≤HS/GED <4 yrs college ≥4 yrs college	7 (6.0) 58 (49.6) 52 (44.4)	4 (8.7) 22 (47.8) 20 (43.5)	2 (5.0) 19 (47.5) 19 (47.5)	1 (3.2) 17 (54.8) 13 (41.9)	0.884
Rank Patrol officer Sergeant/Lieut/Capt Det/Exec/Other	69 (59.0) 20 (17.1) 28 (23.9)	25 (54.4) 10 (21.7) 11 (23.9)	24 (60.0) 4 (10.0) 12 (30.0)	20 (64.5) 6 (19.4) 5 (16.1)	0.464
BMI (Kg/m^2^) Normal (<25.0) Overweight (25–29) Obese (≥30)	18 (15.4) 61 (52.1) 38 (32.5)	10 (21.7) 18 (39.1) 18 (39.1)	5 (12.5) 23 (57.5) 12 (30.0)	3 (9.7) 20 (64.5) 8 (25.8)	0.212
Smoking status Never Former Current	74 (63.8) 32 (27.6) 10 (8.6)	33 (73.3) 12 (26.7) 0 (0.0)	25 (62.5) 12 (30.0) 3 (7.5)	16 (51.6) 8 (25.8) 7 (22.6)	0.015
Sleep quality Good Poor	31 (27.4) 82 (72.6)	15 (34.1) 29 (65.9)	8 (20.5) 31 (79.5)	8 (26.7) 22 (73.3)	0.382
Hypertension No Yes	93 (79.5) 24 (20.5)	36 (78.3) 10 (21.7)	31 (77.5) 9 (22.5)	26 (83.9) 5 (16.1)	0.777
Metabolic Syndrome Yes (≥3 components) No	27 (23.1) 90 (76.9)	12 (26.1) 34 (73.9)	8 (20.0) 32 (80.0)	7 (22.6) 24 (77.4)	0.798
Second job No Yes	81 (69.8) 35 (30.2)	35 (76.1) 11 (23.9)	31 (77.5) 9 (22.5)	15 (50.0) 15 (50.0)	0.023
Workload High Med/Low	78 (68.4) 36 (31.6)	30 (68.2) 14 (31.8)	25 (62.5) 15 (37.5)	23 (76.7) 7 (23.3)	0.451

Results were obtained from ANOVA (continuous variables) and the chi-square test (categorical variables). Good sleep quality ≤ 5 of PSQI and poor sleep quality ≥ 6 of PSQI. Hypertension: SBP of ≥140 mmHg or a DBP of ≥90 mm Hg or self-reported use of antihypertensive medications. Workload: High = very busy, complaints, high crime area; Mod/Low = moderate/low complaint rate, average/low crime. Percent of hours spent on each shift: Day shift (N = 46): mean = 78.7; median = 80.0; range = 44.9–100. Afternoon shift (N = 40): mean = 72.0; median = 76.0; range = 39.0–97.7. Night shift (N = 31): mean = 68.3; median = 69.9; range = 38.5–99.8.

**Table 2 ijerph-21-00439-t002:** Mean change in CRAE (2^nd^–1^st^ follow-up exam) by shiftwork status.

	Day	Afternoon	Night	*p*-Value
Shiftwork (entire Career)	(n = 46)	(n = 40)	(n = 31)	
Model 1	−1.98 ± 1.34	−0.22 ± 1.43	−0.01 ± 1.61	0.563
Model 2	−1.24 ± 1.35	−1.24 ± 1.40	0.20 ± 1.57	0.738
Shiftwork (past year)	(n = 56)	(n = 45)	(n = 10)	
Model 1	−2.24 ± 1.18	0.20 ± 1.31	0.97 ± 2.79	0.307
Model 2	−1.74 ± 1.16	−0.29 ± 1.30	0.36 ± 2.65	0.640
Percent hours on day shift (entire career)	**<70% (n = 90)**	**≥70% (n = 27)**		
Model 1	0.23 ± 0.93	−4.49 ± 1.71		0.017
Model 2	0.56 ± 0.89	−5.57 ± 1.71		0.003

Results are mean ± SE from ANCOVA. Model 1: Adjusted for age. Model 2: Adjusted for age, sex, hypertension, BMI, waist circumference, and change in CRVE (2^nd^–1^st^ follow-up exam). Entire career: Day vs. Afternoon: *p* = 0.998. Day vs. Night: *p* = 0.501. Afternoon vs. Night: *p* = 0.489. Past year: Day vs. Afternoon: *p* = 0.426. Day vs. Night: *p* = 0.473. Afternoon vs. Night: *p* = 0.826.

**Table 3 ijerph-21-00439-t003:** Mean change in CRAE (2^nd^–1^st^ follow-up exam) by shiftwork status stratified by sex and police rank.

	Women (n = 32)			Men (n = 85)		
	Day	Afternoon/Night	*p*	Day	Afternoon/Night	*p*
Shiftwork (entire career)	(n = 20)	(n = 12)		(n = 26)	(n = 59)	
Model 1	−2.71 ± 1.98	3.20 ± 2.58	0.084	−1.41 ± 1.80	−0.81 ± 1.19	0.784
Model 2	−2.20 ± 1.83	2.35 ± 2.40	0.156	−0.45 ± 1.75	−1.23 ± 1.14	0.717
Shiftwork (past year)	(n = 23)	(n = 8)		(n = 33)	(n = 47)	
Model 1	−3.09 ± 1.58	3.83 ± 2.79	0.045	−1.49 ± 1.59	−0.37 ± 1.33	0.593
Model 2	−2.65 ± 1.53	2.58 ± 2.75	0.125	−0.92 ± 1.50	−0.77 ± 1.25	0.939
Percent hours on day shift (entire career)	**<70% (n = 16)**	**≥70% (n = 16**)		**<70% (n = 74)**	**≥70% (n = 11)**	
Model 1	1.76 ± 2.27	−2.75 ± 2.27	0.178	−0.13 ± 1.02	−6.83 ± 2.66	0.021
Model 2	2.02 ± 2.02	−3.00 ± 2.02	0.099	−0.08 ± 0.96	−7.13 ± 2.51	0.011
	**Patrol Officers (n = 69)**			**Other Officers (n = 48)**		
	**Day**	**Afternoon/Night**	** *p* **	**Day**	**Afternoon/Night**	** *p* **
Shiftwork (entire career)	(n = 25)	(n = 44)		(n = 21)	(n = 27)	
Model 1	−2.28 ± 1.83	−0.58 ± 1.38	0.462	−1.16 ± 2.04	0.24 ± 1.78	0.624
Model 2	−2.21 ± 1.97	−0.61 ± 1.44	0.535	−0.58 ± 2.01	−0.21 ± 1.73	0.898
Shiftwork (past year)	(n = 30)	(n = 38)		(n = 26)	(n = 17)	
Model 1	−1.94 ± 1.67	−0.37 ± 1.48	0.485	−2.46 ± 1.62	1.73 ± 2.01	0.114
Model 2	−2.08 ± 1.74	−0.26 ± 1.53	0.459	−1.61 ± 1.46	0.44 ± 1.83	0.402
Percent hours on day shift (entire career)	**<70% (n = 53)**	**≥70% (n = 16)**		**<70% (n = 37)**	**≥70% (n = 11)**	
Model 1	−0.26 ± 1.24	−4.30 ± 2.27	0.124	0.87 ± 1.42	−4.56 ± 2.66	0.082
Model 2	0.48 ± 1.25	−6.75 ± 2.45	0.015	0.92 ± 1.33	−4.73 ± 2.61	0.072

Results are mean ± SE from ANCOVA. Model 1: Adjusted for age. Model 2: Adjusted for age, sex, hypertension, BMI, waist circumference, and change in CRVE (2^nd^–1^st^ follow-up exam). Sex-stratified results were not adjusted for sex.

**Table 4 ijerph-21-00439-t004:** Mean change in CRVE (2^nd^–1^st^ follow-up exam) by shiftwork status.

	Day	Afternoon	Night	*p*
Shiftwork (entire Career)	(n = 46)	(n = 40)	(n = 31)	
Model 1	−1.62 ± 1.38	2.79 ± 1.48	−0.40 ± 1.66	0.092
Model 2	−1.00 ± 1.36	2.92 ± 1.39	−1.46 ± 1.62	0.060
Shiftwork (past year)	(n = 56)	(n = 45)	(n = 10)	
Model 1	−0.27 ± 1.29	−0.02 ± 1.43	1.83 ± 3.05	0.819
Model 2	0.06 ± 1.23	0.08 ± 1.38	−0.50 ± 2.85	0.982
Percent hours on day shift (entire career)	**<70% (n = 90)**	**≥70% (n = 27)**		
Model 1	−0.26 ± 0.99	1.76 ± 1.82		0.337
Model 2	−0.70 ± 0.95	3.15 ± 1.84		0.078

Results are mean ± SE from ANCOVA. Model 1: Adjusted for age. Model 2: Adjusted for age, sex, WBC count, sleep duration, and change in CRAE (2^nd^–1^st^ follow-up exam). Entire career: Day vs. Afternoon: *p* = 0.051. Day vs. Night: *p* = 0.837. Afternoon vs. Night: *p* = 0.041. Past year: Day vs. Afternoon: *p* = 0.994. Day vs. Night: *p* = 0.859. Afternoon vs. Night: *p* = 0.857.

**Table 5 ijerph-21-00439-t005:** Mean change in CRVE (2^nd^–1^st^ follow-up exam) by shiftwork status stratified by sex and police rank.

	Women (n = 32)			Men (n = 85)		
	Day	Afternoon/Night	*p*	Day	Afternoon/Night	*p*
Shiftwork (entire career)	(n = 20)	(n = 12)		(n = 26)	(n = 59)	
Model 1	0.42 ± 2.08	4.59 ± 2.71	0.240	−3.28 ± 1.83	0.78 ± 1.21	0.070
Model 2	2.35 ± 1.53	1.28 ± 2.14	0.702	−2.73 ± 1.75	0.52 ± 1.12	0.128
Shiftwork (past year)	(n = 23)	(n = 8)		(n = 33)	(n = 47)	
Model 1	1.67 ± 2.02	3.01 ± 3.56	0.754	−1.88 ± 1.65	0.04 ± 1.38	0.376
Model 2	3.66 ± 1.34	−3.45 ± 2.70	0.037	−1.47 ± 1.52	−0.24 ± 1.25	0.536
Percent hours on day shift (entire career)	**<70% (n = 16)**	**≥70% (n = 16)**		**<70% (n = 74)**	**≥70% (n = 11)**	
Model 1	1.19 ± 2.38	2.78 ± 2.38	0.646	−0.51 ± 1.10	−0.14 ± 2.86	0.906
Model 2	0.53 ± 1.71	3.31 ± 1.65	0.271	−0.85 ± 1.01	2.21 ± 2.70	0.297
	**Patrol Officers (n = 69)**			**Other officers (n = 48)**		
	**Day**	**Afternoon/Night**	** *p* **	**Day**	**Afternoon/Night**	** *p* **
Shiftwork (entire career)	(n = 25)	(n = 44)		(n = 21)	(n = 27)	
Model 1	−2.26 ± 1.95	0.28 ± 1.47	0.302	−0.27 ± 2.02	2.76 ± 1.76	0.285
Model 2	−1.58 ± 2.13	−0.17 ± 1.53	0.615	−0.16 ± 1.92	2.67 ± 1.66	0.304
Shiftwork (past year)	(n = 30)	(n = 38)		(n = 26)	(n = 17)	
Model 1	−0.62 ± 1.80	−1.02 ± 1.60	0.866	0.27 ± 1.79	3.11 ± 2.22	0.327
Model 2	−0.06 ± 1.86	−1.53 ± 1.62	0.579	0.96 ± 1.65	2.06 ± 2.07	0.688
Percent hours on day shift (entire career)	**<70% (n = 53)**	**≥70% (n = 16)**		**<70% (n = 37)**	**≥70% (n = 11)**	
Model 1	−1.61 ± 1.33	2.57 ± 2.43	0.136	1.57 ± 1.47	0.98 ± 2.74	0.853
Model 2	−2.32 ± 1.32	4.56 ± 2.56	0.027	1.16 ± 1.39	2.36 ± 2.76	0.712

Results are mean ± SE from ANCOVA. Model 1: Adjusted for age. Model 2: Adjusted for age, sex, WBC count, sleep duration, and change in CRAE (2^nd^–1^st^ follow-up exam). Sex-stratified results were not adjusted for sex.

## Data Availability

The SAS data used to support the findings of this Buffalo Cardio-Metabolic Occupational Police Stress (BCOPS) study have not been made available because of the sensitive nature of the health-related outcomes and the potential for participant re-identification.

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
