# Peer review of "Impact of Shiftwork on Retinal Vasculature Diameters over a 5-Year Period: A Preliminary Investigation Using the BCOPS Study Data"

_ijerph, 2024, doi:10.3390/ijerph21040439_

Round 1

Reviewer 1 Report

Comments and Suggestions for Authors

This manuscript reports significantly greater adverse changes in central retinal venular equivalent of patrol officer shift workers who “who worked ≥70% day, compared to those who worked <70% day”. The authors found results surprising since a solid body of evidence shows that shift-work usually associates with greater health risks.

Results are novel and interesting. Discussion can be improved, however, and there are several issues that should be solved before manuscript can be recommended for publication:

1. Several factors can cause a difference in adaption to shift-work and its health consequences. Major factors are chronotype and an exact starting time of shift-work (e.g.,doi: 10.1371/journal.pone.0053379). It is a pity that the authors left off-scope individual circadian preferences of participants (chronotype). Evening types may adapt faster to changing in working schedules including shifts, but they also may be less resistant to changing working regimen and have lower endurance with increase shift work duration / experience (doi.org/10.1080/07420528.2023.2256839).

2. Examination of Table 1 raise several questions: Shift-workers in the “Night” groups are borderline significantly (p<0.1) younger, have fewer “years of service” experience, and also borderline lower body fat percentage. Therefore, it is interesting to know whether adjustment for extra-factor of “years of service ” would impact results. It was shown recently that duration of rotating sojourn-work experience had larger impact on metabolic health than mean age, and also may impact on endurance of certain chronotypes (doi.org/10.1080/07420528.2023.2256839).

3. Another putative factor that could have biased main findings is regularity of shift-work. Even though workers with <70% day had more daywork, it could be that their schedule was more consistent with less frequent changes during years of service, or during the year before examination.

4. One more co-factor that can be discussed is coffee / tea consumption that expected to be higher in night shift-workers and may have different impact on visual function in short-term vs long-term perspective, e.g. long-term effects can be protective.

Author Response

Dear reviewer, thank you for your careful review of our manuscript. Please see the attached document.

Reviewer 2 Report

Comments and Suggestions for Authors

-How do you think the results affected by the fact that your final sample was about 1/3 of the beginning sample (~70% lost to follow up) ?

-Multiple analyzes are performed looking for a statistically significant result. There is the possibility the results be overestimated due to the small sample size. Especially in the ANCOVA analysis one of the study groups includes 8 subjects reducing the power of the model. The results are uncertain and adjustment cannot be done for 6 variables (model 2, table 3).

-Have you thought about using the arteriovenous ratio (AVR), as a more reliable index than CRAE and CRVE, for yours statistical analysis?

-Causality is not established from the study design and data analysis. I would suggest removing the sentence: “In addition, the longitudinal study design allows us to infer causality” from the paragraph: “4.3. Limitations and Strengths”

Reviewer 3 Report

Comments and Suggestions for Authors

This is an interesting and important study addessing the impact of shiftwork on various retinal parameters. Overall, the manuscript is well-written and my comments are mostly minor.

1) The authors used a cutoff-point of 70% for percentage of hours worked on the day shift. This value is somewhat arbitrary and needs to be mentioned as a limitation in the discussion section.

2) There are many p-values reported.  Adjustment should be done for multiplicity or else mentioned as a limitation.

3) The authors have take great care to adjust for important outcome related covariates.  However, as groups were not randomized, there possibly could be imbalances with respect to unmeasured confounders.  Please convince this reviewer otherwise.

4) Given the small sample size, I would suggest mentioning in the title that this is a pilot study.

Comments on the Quality of English Language

Needs minor English language editing.

Author Response

Thank you for your careful review of our manuscript. Please see the attachment.
